# Unexpected Interacting Effects of Physical (Radiation) and Chemical (Bisphenol A) Treatments on Male Reproductive Functions in Mice

**DOI:** 10.3390/ijms222111808

**Published:** 2021-10-30

**Authors:** Margaux Wieckowski, Stéphanie Ranga, Delphine Moison, Sébastien Messiaen, Sonia Abdallah, Sylvie Granon, René Habert, Virginie Rouiller-Fabre, Gabriel Livera, Marie-Justine Guerquin

**Affiliations:** 1Laboratory of Development of the Gonads, UMR-008 Genetic Stability Stem Cells and Radiations, Université de Paris, 92265 Fontenay-aux-Roses, France; margaux.wieckowski@ephe.psl.eu (M.W.); ranga.stephanie@yahoo.fr (S.R.); delphine.moison@cea.fr (D.M.); sebastien.messiaen@cea.fr (S.M.); sonia.abdallah06@gmail.com (S.A.); r.habert@orange.fr (R.H.); virginie.rouiller-fabre@cea.fr (V.R.-F.); 2Université Paris Saclay, CEA/DRF/IBFJ/IRCM, 98 Route du Panorama, 92265 Fontenay-aux-Roses, France; 3Neuroscience Paris-Saclay Institute (Neuro-PSI), CNRS UMR 9197, Paris-Sud University, 91400 Saclay, France; sylvie.granon@universite-paris-saclay.fr; 4Paris-Saclay University, 91405 Orsay, France

**Keywords:** endocrine disruptors, reproduction, germ cells, Leydig cells, γ-ray, Bisphenol A

## Abstract

For decades, numerous chemical pollutants have been described to interfere with endogenous hormone metabolism/signaling altering reproductive functions. Among these endocrine disrupting substances, Bisphenol A (BPA), a widely used compound, is known to negatively impact germ and somatic cells in the testis. Physical agents, such as ionizing radiation, were also described to perturb spermatogenesis. Despite the fact that we are constantly exposed to numerous environmental chemical and physical compounds, very few studies explore the impact of combined exposure to chemical and physical pollutants on reproductive health. The aim of this study was to describe the impact of fetal co-exposure to BPA and IR on testicular function in mice. We exposed pregnant mice to 10 µM BPA (corresponding to 0.5 mg/kg/day) in drinking water from 10.5 dpc until birth, and we irradiated mice with 0.2 Gy (γ-ray, RAD) at 12.5 days post-conception. Co-exposure to BPA and γ-ray induces DNA damage in fetal germ cells in an additive manner, leading to a long-lasting decrease in germ cell abundance. We also observed significant alteration of adult steroidogenesis by RAD exposure independently of the BPA exposure. This is illustrated by the downregulation of steroidogenic genes and the decrease of the number of adult Leydig cells. As a consequence, courtship behavior is modified, and male ultrasonic vocalizations associated with courtship decreased. In conclusion, this study provides evidence for the importance of broadening the concept of endocrine disruptors to include physical agents, leading to a reevaluation of risk management and regulatory decisions.

## 1. Introduction

Exposure to chemical and physical pollutants in the environment has broadly increased in industrialized countries in the past 100 years. We are constantly exposed to a variety of chemical and physical compounds that may affect reproductive function. Spermatogenesis and steroidogenesis, the two main testicular functions, have been reported to be altered by numerous pollutants, especially during the establishment of these functions in the developing gonad. For this reason, fetal life is a uniquely sensitive window for pollutant exposure. The susceptibility of fetal germ cells to DNA damage and oxidative stress makes them a privileged target for radiation and chemotherapeutic drugs, and the adverse effects of radiation on gonocytes (apoptosis and mutagenicity) have been well known for more than half a century [1,2,3,4,5]. Ionizing radiation (IR) comes from different natural (radon gas, cosmic rays) and anthropogenic (diagnostic imaging, radiotherapy, nuclear accidents or nuclear weapon testing) sources. Exposure to artificial sources of ionizing radiation is constantly increasing, mainly due to the medical use of radiation. In France, between 2014–2019, the average dose of exposure *per capita* was estimated by IRSN to be 4.5–6.5 mSv/yr [6]. IR affects human fertility, mostly impacting proliferative germ cells (i.e., gonocytes and spermatogonia) [1,3,7]. Moreover, postnatal irradiation also seems to have an impact on testicular somatic cells, particularly Leydig cells, as a decrease in androgen production was reported after irradiation [8,9,10,11,12,13,14]. Other exogenous substances, such as numerous plasticizers, pesticides and drugs, are more prone to interfering directly with the endocrine system, altering fetal Leydig cell function and androgen production [15]. Bisphenol A (BPA) is a compound used in the manufacture of many plastics, epoxy and polyester resins, and due to the pervasive presence of BPA in our environment, humans are continuously exposed to it. Thus, human biological fluids (urine, blood, serum, amniotic fluid, etc.) from most subjects contain detectable levels of BPA metabolites [16,17]. The mean concentrations of BPA metabolites in blood samples were reported to be in the range of 1–10 nM [18]. As BPA is defined as a xenoestrogen and has weak estrogenic and anti-androgenic properties, it is considered to belong to the huge family of endocrine disrupting compounds [19]. Numerous epidemiological and experimental studies have demonstrated that BPA exposure can alter steroidogenesis, leading to male reproductive disorders [19,20,21]. Positive correlations of cryptorchidism [22] and decreased anoscrotal/anogenital distance [23,24] with BPA levels in plasma or cord blood were observed, suggesting that this chemical has an antiandrogenic effect during pregnancy. This hypothesis is strengthened by the results of experimental studies in rodents or human models, which show numerous testicular impairments in androgen production and gametogenesis, leading to a decrease in sperm count after prenatal exposure to BPA [21,25,26]. Additionally, BPA exposure has recently been proposed to induce numerous types of DNA damage (DNA oxidation, adducts and double strand breaks) and genotoxicity [27]. For this reason, BPA toxicity on reproductive function could be explained not only by its endocrine disrupting activity but also by its direct damaging effects in somatic and germ cells.

The assessment of pollutants is mostly based on the evaluation of a single pollutant or a combination of chemicals with common features and modes of action. In this case, the adverse effect of these substances can be mainly extrapolated as a result of additive concentration/dose effects [28,29]. The risk assessment of compounds with different modes of action is more complex and difficult to predict. Few studies have evaluated the combination of physical and chemical pollutants and their effects on health, particularly reproductive functions, despite their systematic cooccurrence in our environment [30,31,32,33,34,35,36,37,38]. 

In this context, we aim to investigate the long-term effects of combined fetal exposure to two well-known environmental toxicants, γ-rays and BPA, on male reproduction. Even though the cooccurrence of γ-rays and BPA in the environment does not correspond to a specific real-life situation, these agents are chosen as models of DNA damage inducers and mutagens and endocrine disrupting compounds, respectively. The type of synergism between these pollutants was studied, focusing on short- and long-term defects in spermatogenesis, Leydig cell function and reproductive behavior in male mice. Four experimental groups were studied, and they were nonirradiated and non-BPA-exposed control mice (CTL); nonirradiated and BPA-exposed mice (BPA); irradiated and non-BPA-exposed mice (RAD); irradiated and BPA-exposed mice (BPA_RAD). For BPA exposure, pregnant mice were exposed to 10 µM BPA (corresponding to 0.5 mg/kg/day) in drinking water from 10.5 dpc to 18.5 dpc. For RAD exposure, 12.5 dpc pregnant mice were irradiated with a single whole-body dose (0.2 Gy). We observed that co-exposure to BPA and γ-ray induced DNA damage in fetal germ cells in an additive manner, leading to a long-lasting decrease in germ cell abundance. Transcriptomic analyses revealed that gene expression was strongly impacted in the adult testis following the co-exposition, while exposure to BPA or radiation alone had little lasting effect on gene expression. Unexpectedly, we also observed the alteration of the adult steroidogenesis function after fetal γ-ray exposure, independent of BPA exposure. This alteration is illustrated by the downregulation of steroidogenic genes and the decrease in the number of adult Leydig cells. As a consequence, courtship behavior was modified, and male ultrasonic vocalizations associated with courtship decreased. The understanding of the interaction between pollutants with distinct modes of action could help with risk management and regulatory decision-making concerning environmental pollutants.

## 2. Results

### 2.1. Synergistic Effects of BPA and RAD on DSBs and Consequences for Germ Cell Proliferation and Viability

After BPA and/or ionizing radiation (RAD) exposure, DNA double-strand breaks (DSBs) were determined indirectly through the detection of γH2AX in 12.5 dpc germ cells. The number of γH2AX foci *per* nucleus section in germ cells (DDX4 positive cells) was determined under each condition at 6 h postirradiation. Low doses of RAD or BPA alone did not significantly increase the number of γH2AX foci in comparison to the control treatment (Figure 1A). The combination of BPA and RAD (BPA_RAD) exposure increased the number of γH2AX foci significantly and in an additive manner (Figure 1A). To assess the consequences of this damage on germ cell proliferation, we detected the phosphorylation of histone H3 (pH3), a mitotic cell marker in TRA98 positive cells. A significant decrease in pH3-positive germ cells was observed in all treatment conditions (BPA, RAD alone or in combination) in comparison to the control (CTL) condition. No additive effect of BPA and RAD was observed for germ cell proliferation (Figure 1B). BPA and/or RAD exposure during fetal germ cell proliferation led to a significant decrease in the number of germ cells at 15.5 dpc (Figure 1C). No difference in germ cell count between the control and BPA or RAD treatment was observed at 18.5 dpc or 1 dpp. The combination of BPA and RAD induces a persistent decrease in the number of germ cells until 1 dpp (Figure 1C). In conclusion, cumulative DSBs were rapidly observed after combined exposure to BPA and RAD during the proliferative and pluripotent stages, durably impacting the germ cell proliferation and survival and inducing lasting germ cell loss.

### 2.2. BPA and RAD Induced Long-Term Transcriptional Alterations in the Testis

Using an Affymetrix mouse genome expression microarray (MTA.1), we analyzed gene expression in fetal (18.5 dpc) and adult (3 months) testes from mice, exposed to BPA and/or RAD in utero (Figure 2). We identified significant changes in gene expression in treated testes (BPA, RAD alone or in combination) in comparison to the controls. Only genes expressed with a *p* value ≤ 0.05 and a ǀLogFCǀ ≥ 0.5 were considered significantly differentially expressed. At 18.5 dpc, we observed 351 and 427 probe sets that were significantly changed after BPA or RAD treatment, respectively (Figure 2A). Twenty-five to 30% of these genes were common between BPA and RAD conditions; the genes were preferentially enriched in cell death/apoptosis, stress and transcriptional regulation functions and included *Foxa1, Fos, Fosb, Klf4, Atf3, Nr4a1 and Egr1* (Appendix A). At 18.5 dpc, the combination of BPA and RAD did not increase the number of DEGs (Figure 2A). These DEGs (389 genes) were mostly shared with those induced by BPA or RAD (approximately 75%) treatment (Figure 2A). At 3 months, although few DEGs were observed after BPA (39 genes) or RAD (136 genes) treatment alone, we observed 975 DEGs in the BPA_RAD condition (Figure 2A,B). The combination of BPA and RAD exposure in utero enhances the transcriptional response in adulthood in terms of the number of DEGs and logFC (Figure 2B). Gene-by-gene expression analyses after exposure to both BPA and RAD, in comparison to BPA or RAD alone at 18.5dpc and at 3 months, showed non-predicted interactions (i.e., additive interaction) between treatments (Table 1).

Three types of interactions were considered, including additivity, antagonism and potentiation. At 18.5 dpc, we observed an antagonistic effect between RAD and BPA on the expression of two-thirds of the genes. These genes were not disproportionately associated with specific pathways. At 3 months, the alteration of gene expression was mostly potentiated by the combination of BPA and RAD, as observed for *Star, Hsd3b1* and *Hsd17b3* (Table 1, Figure 2B, Appendix A). As observed under RAD conditions, differentially expressed genes (DEGs) were mostly downregulated in BPA_RAD co-exposed testes, and the repression was enhanced in comparison to RAD treatment alone for some of these genes linked to *steroid metabolic and synthetic process* pathways (Figure 2B,C, Appendix A). Moreover, using the online tool EnrichR [39], we demonstrated that downregulated genes that were common between the RAD and BPA_RAD conditions had predicted association with *XY female disease* (*p* value: 5.3 × 10^−5^) and *androgen insensitivity syndrome* (*p* value: 7.3 × 10^−4^), as defined by GeneRIF datasets and the co-expression matrix from ARCHS4. Using these two gene sets obtained from the *XY female disease* and *androgen insensitivity syndrome* libraries in the *Rare Diseases GeneRIF ARCHS4 Predictions* database, we observed a globally significant downregulation of these genes in the RAD and BPA_RAD conditions in comparison to other genes chosen by simple random sampling (Figure 2D, Appendix A). In conclusion, combined exposure of BPA and RAD during fetal life leads to long-lasting transcriptional deregulation of genes related to steroidogenesis. 

### 2.3. BPA and RAD Act Synergistically to Induce Gonocyte-to-Spermatogonia Transition

Gene ontology (GO) analysis of the DEGs in BPA, RAD and BPA_RAD treated 18.5 dpc testes revealed that the *methylation involved in gamete generation germ cell* process (GO:0043046) was enriched in RAD and BPA_RAD. Genes linked to this process, such as *Dnmt3l* and *Tdrd9,* were mostly downregulated in the RAD and BPA_RAD groups (Appendix A). These genes are involved in DNA methylation and transposon silencing during the quiescence phase, and downregulation of these genes occurs during the gonocyte-to-spermatogonia transition [40]. As DNA demethylation and resumption of mitosis are hallmarks of the gonocyte-to-spermatogonia transition, we performed 5mC (Figure 3A) and KI67 (Figure 3B) immunodetection on 18.5 dpc exposed testes. Under control conditions, most germ cells presented characteristics of gonocytes, with 5mC-positive and KI67-negative nuclei and were localized in the center of the seminiferous cords. No significant modification of germ cell DNA methylation or proliferation was observed after single BPA or RAD exposure (Figure 3). After BPA_RAD treatment, however, we observed a significant increase in nonmethylated germ cells (Figure 3A). These cells were localized at the basement membrane and had features typical of proliferative cells. We also observed an increase in the number of KI67-positive germ cells in testes exposed to BPA and RAD compared to control conditions (Figure 3B). In conclusion, combined exposure to BPA and RAD during fetal life perturbs quiescence resumption and germ cell differentiation in the perinatal testes, which could affect spermatogenesis in adulthood.

### 2.4. BPA and RAD Act Synergistically to Decrease Spermatogonia and Sperm Contents

To characterize the impact of in utero exposure to BPA and/or RAD on spermatogenesis in adulthood, we examined the pool of spermatogonia in adult testes. Flow cytometry analyses indicated a decrease in the percentage of spermatogonia after RAD treatment alone or combined with BPA (Figure 4A). A similar decrease was observed in 6-month-old testes exposed with RAD alone or in combination with BPA in utero. The combined exposure did not significantly enhance this decrease. BPA exposure alone did not impact the spermatogonia content at 3 months, but a decreasing tendency was observed at 6 months (Figure 4A). These results were confirmed by immunodetection of PLZF, a marker of undifferentiated spermatogonia, in 6-month-old testes and quantification of the number of spermatogonia per basement membrane area (mm^2^). After BPA and/or RAD exposure, we observed a significant decrease in the number of spermatogonia compared to that in the control testes (Figure 4B). 

We did not observe any additive effect or potentiation between BPA and RAD treatment on the density of spermatogonia. Consequently, sperm production was significantly reduced only in the BPA_RAD condition, although we observed a tendency towards a decrease after treatment with BPA or RAD alone (Figure 4C). No modification of the litter size was observed between conditions suggesting that BPA and/or RAD have no impact on sperm maturation (Appendix A). In conclusion, BPA and RAD act synergistically on fetal germ cells, inducing a reduction in the pool of undifferentiated spermatogonia and sperm content.

### 2.5. RAD Fetal Exposure Alone Decreases Androgen Production and the Number of Leydig Cells

After fetal irradiation (RAD and BPA_RAD), we observed a global downregulation of genes linked to *steroid metabolic and synthetic processes* in adult testes. For this reason, we measured serum testosterone levels in adult mice (3–5 months) by radioimmunoassay. We observed a trend towards a reduction in testosterone levels, especially under RAD conditions (Figure 5A). The addition of BPA did not enhance this tendency. Microarray analyses from adult testes suggested an alteration of adult Leydig cells induced by ionizing radiation exposure during fetal development. Indeed, we observed a significant decrease in the expression of adult Leydig cell markers, such as *Hsd3b6* and *Hsd17b3*, in irradiated testes with or without BPA (Appendix A). Moreover, we quantified the Leydig cells in adult testes by CYP11A1 immunodetection. We observed a significant decrease in the number of Leydig cells in RAD-exposed testes in the absence of BPA, and the addition of BPA antagonized this decrease (Figure 5). No effect of BPA and/or RAD exposure on the number of fetal Leydig cells in neonatal (1 dpp) testes was observed (Appendix A). In conclusion, the number of adult Leydig cells decreased after in utero exposure to RAD. This reduction could affect testosterone production and the regulation of steroidogenesis.

### 2.6. RAD Fetal Exposure Decreases the Emission of Male Mouse Ultrasonic Vocalizations during Courtship

Brain masculinization, which induces sexual dimorphic behavior and the emission of male mouse ultrasonic vocalizations (USVs), is under androgenic control [41]. Therefore, male USVs act as attractive sexual signals to female mice and stimulate reproduction. In utero ionizing radiation exposure decreased the Leydig cell number. Thus, we conducted a female partner preference test with in utero BPA- and/or RAD-exposed or unexposed males. A sexually receptive female was placed in one arm of an X maze that offered access to two males from different experimental groups (CTL, BPA, RAD or BPA_RAD). 

The time spent in each arm of the maze was measured to determine the female preference for a male (Figure 6A). No significant female preference was observed in this test (chi-square test, *n* = 5–7), but this analysis showed a tendency for females to prefer nonirradiated males. Indeed, when we compared CTL males with RAD or BPA_RAD males or BPA males with RAD males, we observed that females spent more time with the CTL or BPA males, respectively (Figure 6B). During the partner preference test, we recorded and counted USVs (Figure 6A,D–F). We observed no significant difference in the total number of USVs emitted by males in utero exposed to BPA, RAD or BPA_RAD (Figure 6D). However, we observed a significant decrease in some types of calls involved in courtship in males exposed to BPA and/or RAD (Figure 6E,F). We observed a significant decrease of *Harmonic* songs in all treated groups (BPA, RAD and BPA_RAD males) (Figure 6E,F) compared with CTL males. Moreover, we observed a significant decrease in the emission of *one-jump*, *multi-jump* and *short* USVs by RAD males. *Multi-jump* and *short* USVs were also emitted less often by BPA males and BPA_RAD males (Figure 6E,F). In conclusion, we observed long lasting changes on male reproductive behavior after in utero RAD exposure.

## 3. Discussion

The present study characterizes for the first time the synergistic effects of simultaneous fetal exposure to bisphenol A and ionizing radiation on testicular functions (spermatogenesis and steroidogenesis). Fetal development is a specific window sensitive to numerous pollutants because testicular development during this period is crucial for subsequent male reproductive function. Steroidogenesis takes place during fetal development, and the pool of adult Leydig cells depends on fetal fetal precursor cells. In addition, the first steps of germ cell differentiation begin during fetal development. Germ cell loss of pluripotency, acquisition of gametogenesis competency, orientation toward spermatogenesis and quiescence occurs during fetal life. 

In mammalian fetal testes, germ cells are cells with the greatest sensitivity to ionizing radiation (RAD) [3,42,43]. Doses as low as 0.1 Gy during germ cell proliferation cause DNA double strand breaks (DSBs) and induces cell death by apoptosis [3]. Due to its ability to produce oxidative stress, bisphenol A (BPA) exposure also induces DSBs in germ cells [44,45]. Thus, fetal germ cells appear to be the main direct target of these genotoxic agents, and BPA or RAD exposure alone could impact fetal germ cells. However, low doses of BPA or IR have no consequence on the pool of germ cells at birth despite inducing a significant decrease in germ cell numbers at 15.5 dpc. The rescue of this phenotype could be explained by a putative delay/lack of quiescence entry in some cells after RAD or BPA exposure. The combination of RAD and BPA during the mitotic phase induces cumulative DSBs in male germ cells, leading to cell cycle checkpoint arrest and persistent germ cell loss during fetal life. In addition, alteration of germ cell differentiation was observed, with the early presence of mitotic germ cells with features of spermatogonia at the end in testes simultaneously exposed to BPA and RAD. During the process of male germ cell differentiation, the maintenance of mitotic arrest during the quiescence phase promotes global DNA methylation and prevents retrotransposon reactivation, precocious germ cell differentiation and/or meiosis initiation [40,46,47]. DNA methylation and key actors of this process, such as Dnmt3L, are essential for the maintenance of the quiescence phase [48]. For this reason, the downregulation of genes linked to DNA methylation and the presence of mitotic germ cells is expected, but it is not clear whether the loss of DNA methylation in germ cells is a cause or a consequence of this observed defect in germ cell differentiation. Moreover, how BPA and RAD exposure during the mitotic phase and subsequent DNA damage and oxidative stress could interfere with germ cell mitotic arrest remains an open question. Extrinsic (Notch and TGFb signaling, retinoic acid metabolism, etc.) and intrinsic (metabolic status, epigenetic modifications, Nanos2 expression, etc.) factors control mitotic arrest in germ cells [40,48,49,50,51]. Interestingly, ex vivo exposure of fetal mouse testes to 100 µM BPA increases the testicular expression of Notch proteins [52]. We speculate that the transient activation of Notch1 signaling after BPA exposure promotes the germinal exit of quiescence [46,53]. Delayed mitotic arrest and/or precocious exit of quiescence lead to germ cell apoptosis during fetal life and reduced the pool of spermatogonial stem cells in adulthood [40,48,49,51]. This could explain the significant and persistent loss of the germline pool and consequent adverse effects on sperm production observed in the testes of adult mice after in utero exposure to BPA and RAD. The impact of combined exposure to BPA and X-rays on postnatal male germ cells in pubescent and adult rat testes has also been described [33]. The authors of this study also observed a cumulative effect on DNA damage and a decrease in sperm products, notably in pubescent testes, that was certainly due to the susceptibility of spermatogonia to DNA damage. This suggests that susceptibility to DNA damage is a conserved feature in the male germline both during fetal and post-natal life.

In addition to spermatogenesis deficiency, the combination of BPA with RAD during fetal life enhances the testicular transcriptional response in adulthood. This is illustrated by the global downregulation of genes linked to steroidogenesis and Leydig cells in adult BPA_RAD testes. This suggests global defects in Leydig cell and androgen production after BPA and RAD exposure explaining the *androgen insensitivity* and *XY female disease* transcriptomic signatures. Whole-testis transcriptomic analyses do not allow the distinction between putative impacts of BPA and RAD exposure on gene expression or on Leydig cell number and cell survival. However, we showed a significant reduction in the number of Leydig cells in adult testes only after fetal RAD exposure alone, although there was no decrease in the Leydig cell number in the neonatal testes. Adult Leydig cells arise from multiple progenitors, such as fetal Leydig cells, peritubular myoid cells and vascular pericytes, during puberty [54,55]. Analysis of the expression of distinct markers of adult and fetal Leydig cells such as *Hsd3b6* or *Cyp26b1,* respectively [56,57], in adult testes suggests a common dysfunction of adult and “fetal” Leydig cells. Moreover, the proliferation and differentiation of adult Leydig cells are regulated by multiple testicular and pituitary growth factors and hormones [58]. For these reasons, it is difficult to define whether alterations in adult Leydig cells originate directly from changes in fetal testicular precursors. Unexpectedly, combination with BPA seems to restore the number of Leydig cells after RAD exposure. These antagonistic effects of fetal BPA and RAD exposure on Leydig cell numbers could be explained by the mitogenic effect of BPA on Leydig cells, as observed in the testes of adult rats exposed to BPA during fetal live or in organ culture of fetal mouse testes [52,59]. The lack of a visible impact of simultaneous exposure to BPA and RAD on Leydig cell numbers, despite the global downregulation of steroidogenic genes, suggests an effect of these compounds on the regulation of steroidogenic gene expression. Despite a decrease in Leydig cell number and/or downregulation of genes linked to steroidogenesis after in utero irradiation, no significant decrease in plasma testosterone levels was observed. We could not exclude a compensatory and transient elevation of LH that could increase testosterone production. Despite the lack of significant alteration of plasmatic testosterone levels in irradiated mice (with or without BPA), we observed a significant decrease in some ultrasonic vocalizations (USVs) emitted by these males. Male USVs are highly variable and depend upon social context (such as courtship) and species [60,61]. Male vocalizations are attracting for females [62,63]. This positive phonotaxis depends on the type of call and the production of complex syllables, such as *multi-jump* and *harmonic* USVs, which are more attractive than simple syllables [64]. These complex syllables are often associated with the male–female context, notably during courtship [60,65], and interestingly, we observed a decrease in complex calls rather than in simple calls, notably in RAD males. Moreover, using the partner preference test, we observed a less pronounced female interest in irradiated males. Male USV emission is controlled by a central circuit in which neurons express androgen and estrogen receptors [41,66]. For this reason, we could not exclude transient changes in steroidogenesis during the perinatal period. Taken together, our results suggest that ionizing radiation exposure during fetal life on Leydig cells and steroidogenesis likely impacts male courtship vocalization and, by extension, courtship behavior. Even though fetal Leydig cells are more radioresistant than fetal germ cells [67], this work shows a more pronounced long-term impact of fetal radiation exposure on steroidogenesis and adult Leydig cells than on spermatogenesis. Several studies have demonstrated the negative impact of postnatal irradiation at doses higher than 1 Gy on human and rodent Leydig cells and androgen production [8,9,11,12,13,14]. To our knowledge, very few studies have examined the long-term effect of testicular exposure to radiation in utero on adult Leydig cells [43,67,68,69], and most of these studies have reported no obvious effect. In this study, we provide the first evidence that fetal exposure to radiation has an endocrine disrupting effect independent of direct action on estrogen and androgen receptors. In summary, combined exposure to low doses of physical and chemical agents can enhance the harmful effects on spermatogenesis. However, the simultaneous action of irradiation and BPA on steroidogenesis is unexpected. In this study, we observed that ionizing radiation acts as an endocrine disrupting agent and that exposure to BPA, a well-known endocrine disruptor, seems to reverse this action. While this specific finding is obviously restricted to the BPA and RAD doses used in this study, this work nevertheless demonstrates the complexity of extrapolating the interaction between BPA and RAD in adult steroidogenesis. Reproductive health is constantly affected by numerous environmental pollutants, and risks of exposure to complex compound mixtures in the environment are priorities for human and environmental health organizations. Promoting the risk assessment of mixtures of chemicals with a common mode of action is the most effective way to characterize adverse effects based on the dose/concentration approach. In reality, multiple stressor effects are not necessarily a simple sum of the effects of the individual compounds but also include synergistic, antagonistic and nonpredictable effects, especially in a physiological or long-term context. This study provides new evidence of the complexity of understanding the interaction between a well-known endocrine disrupting compound, BPA, and a well-known genotoxic stressor, ionizing radiation, on testicular function. This is notable because the impact of one pollutant could not be restricted to one mode of action. For example, BPA exhibits multiple modes of action (endocrine disruption, genotoxic stress, etc.), and each depends on the dose, the organism and the cellular target studied. In this study, we observed that ionizing radiation, used as a DNA damage inducer to represent numerous environmental pollutants, could also be considered an endocrine disruptor despite lacking any direct activity on hormone receptors. Thus, it is urgent to take into account all modes of action of all stressors for risk management and regulatory decision-making concerning environmental pollutants, particularly for chemicals with endocrine disrupting activity such as BPA. Of course, the assessment of every combination of pollutants would be unrealistic, but we should not ignore the interaction of physical and chemical pollutants that could share similar modes of action (i.e., oxidative and genotoxic stresses). For this reason, these studies might provide support for increased knowledge and inform in silico models. Additionally, our study might prompt the revision of risk management following radiation exposure that is built on numerous models devoid of endocrine disruptors, while the exposition to such molecules is ubiquitous in real life.

## 4. Materials and Methods

### 4.1. BPA and IR Dose Selection after In Vitro Study

To define the lowest doses of BPA and IR for in vivo experiments, we performed a preliminary in vitro study on HeLa human cervical cancer cells (Appendix A). HeLa were cultured in Dulbecco’s modified Eagle’s medium, GlutaMAX™, with low glucose and pyruvate (Merck 21885-025), supplemented with 10% fetal bovine serum and 1% penicillin and grown at 37 °C with 5% CO_2_ in a humidified incubator. After reaching mid-confluence, cells were exposed to IR or BPA. For BPA exposure, medium was supplemented with BPA (from 10^−9^ to 10^−4^ M, >99% purity, Merck) or vehicle (ethanol) for 24 h. In vitro irradiation was performed with a GSR D1 irradiator from GSM Company. It is a self-shielded irradiator with four sources of ^137^Cesium with a total activity of approximately 180.28 TBq in March 2014 that emits gamma rays. Samples were irradiated at different single doses, which were 2.5 and 10 Gy with a dose output of approximately 1.06 Gy/min considering the radioactive decay and 2, 20 and 200 mGy with a dose output of approximately 10.54 mGy/min considering the radioactive decay. Cells were irradiated in a special container in a 6-well plate. Prior to irradiation, dosimetry was performed. As DNA double strand breaks (DSBs) are commonly observed both after IR and BPA exposure [27,70], we quantified the radiation-induced nuclear γ-H2AX foci by immunodetection (Ser139, Millipore 05-636, 1:500) at 6 h after exposure. The doses of 0.2 Gy of IR and 10^−8^ M BPA correspond to the effective doses that induced a significant increase in DSBs compared to the control condition (Appendix A). For this reason, we chose the dose of 0.2 Gy for in vivo exposure. For in vivo BPA exposure, we chose to expose pregnant mice to 10 µM BPA (corresponding to 0.5 mg/kg/day) in drinking water. At the end of exposure (8 days of exposure), the plasma of BPA-treated mice contained 3.3 ± 1.8 10^−8^ M total BPA (corresponding to 7.5 ng/mL by GC-MS/MS detection as previously described) [25]. 

### 4.2. Animal Housing and Experimental Design

All animal procedures described in this study were approved in accordance with the guidelines for the care and use of laboratory animals of the French Ministry of Agriculture (France). Outbred NMRI mice were housed under controlled photoperiod conditions (lights on from 08:00 to 20:00) and were supplied with commercial food and tap water *ad libitum*. Males and females were caged together overnight, and the presence of vaginal plugs was determined the following morning. The day following overnight mating was counted as 0.5 days post conception (dpc). Pregnant mice were randomly divided into four experimental groups, which were nonirradiated and non-BPA-exposed control mice (CTL); nonirradiated and BPA-exposed mice (BPA); irradiated and non-BPA-exposed mice (RAD); irradiated and BPA-exposed mice (BPA_RAD). BPA-exposed pregnant mice were exposed to 10 µM BPA (diluted in 0.1% ethanol) in drinking water from 10.5 days post-coïtum (dpc) to 18.5 dpc. Non-BPA-exposed pregnant mice (ETOH) received drinking water containing 0.1% ethanol. Mice were irradiated at 12.5 dpc with a ^60^Cobalt medical irradiator (Alcyon) with a total activity of approximately 279 TBq in June 2005. Mice were irradiated in a special container with a single whole-body dose (0.2 Gy) with a dose output of 0.02 Gy/min. Prior to irradiation, dosimetry was performed. Nonirradiated mice (SHAM) were manipulated and housed in the same manner as irradiated mice. 

### 4.3. Histology and Immunostaining of Tissue Sections

Pregnant mice were killed by cervical dislocation and the embryos were quickly removed from the uterus for dissection under a binocular microscope. Fetal (12.5, 15.5 and 18.5 dpc), neonatal (1 day postpartum [dpp]) and adult (3 and 6 months) testes were fixed in Bouin’s or formaldehyde fixative. For immunohistochemical staining (KI67, 5mC, PLZF, CYP11A1, HSD3B, KI67, DDX4), sections were submitted to antigen retrieval with citrate buffer (pH 6, KI67; 5mC, CYP11A1, HSD3B, TRA98, KI67, DDX4) or Tris EDTA buffer (pH 9; PLZF). Endogenous peroxidase activity was blocked by incubating sections with 3% hydrogen peroxide for 15 min. Sections were then blocked with 2% horse serum for 30 min and incubated overnight at 4 °C with primary antibodies such as monoclonal mouse anti-5mC (Abcam ab10805, 1:100), monoclonal mouse anti-KI67 (Cell Signaling, 8D5, 1/200), monoclonal rat anti-TRA98 (Abcam ab82527, 1:500), monoclonal mouse anti-MVH/DDX4 (Abcam ab27591; 1:500), polyclonal rabbit anti-PLZF (Santa–Cruz SC28319, 1/50), anti-CYP11A1 (Lsbio LS-C14785,1/200) and anti-HSD3B (TransGenicInc KO607, 1/200). After washing in PBS, the slides were incubated with peroxidase-conjugated appropriate secondary antibodies (ImmPRESS reagent kit, Vector Laboratories, Eurobio) for 30 min at room temperature. Antibodies were revealed with DAB (DAB Substrate Reagent Kit, Vector Laboratories) or VIP Vector (VIP Substrate Reagent Kit, Vector Laboratories).

For immunofluorescence staining (γH2AX, phospho-H3, TRA98, DDX4), sections were submitted to antigen retrieval with citrate buffer (pH 6) and were then blocked in normal horse serum (Impress HRP Reagent Kit MP-7402) or 2% gelatin, 0.05% Tween and 0.2% BSA for one hour before adding antibodies. The primary antibodies used in this study were as follows: monoclonal mouse anti-MVH/DDX4 (Abcam ab27591; 1:500), monoclonal rat anti-TRA98 (Abcam ab82527, 1:500), monoclonal mouse anti-phospho-histone H3 (Ser10, Abcam 9706S, 1:1000) and monoclonal mouse anti-phospho-Histone H2AX (Ser139, Millipore 05-636, 1:500). Specific donkey secondary antibodies were conjugated with either Alexa Fluor 488 or 594 (1:500). Slides were mounted in Vectashield medium. For γH2AX analyses, image acquisition was accomplished with a laser-scanning confocal microscope (confocal Leica TCS SP8), and images were analyzed using ImageJ software. For other analyses, images were acquired using a Leica DM5500 B epifluorescence microscope (Leica Microsystems) equipped with a CoolSNAP HQ2 camera (Photometrics) and ImageJ software (https://imagej.nih.gov/ij/, accessed on 27 October 2021). Images were analyzed with the ImageJ software. 

For germ cell counting, all TRA98-positive cells in each of five sections, equidistantly distributed along the gonad, were counted using Histolab software (Microvision Instruments). The extrapolation of the total cell count was obtained using the Abercrombie formula to correct for any double counting, as previously described [42]. For Leydig cell counts, CYP11A1 (adult testis)- and HSD3B (postnatal testis)-positive cells were counted and normalized to the total surface area of the testis section. Five sections by the testis were counted. For spermatogonial cell counting, PLZF positive cells were counted and reported to the surface area of the tubule section. Thirty sections of seminiferous tubules *per* testis were counted. 

### 4.4. RNA Extraction and Gene Chip Analyses

Total RNA from fetal (18.5 dpc) and adult (3 months) testes was extracted using the Qiagen Rneasy Mini Kit as recommended by the manufacturer. Total RNA concentration and RNA integrity were monitored by electrophoresis (Agilent Bioanalyzer; RNA 6000 Pico Assay). Two pools of fetal and adult gonads from five independent exposures were used for differential expression analyses. Gene expression analysis was conducted using Mouse Transcriptome Assay 1.0 array (Affymetrix) as recommended by the manufacturer. Raw data were generated and controlled with Expression console (Affymetrix) on the GenomIC’s platform (Institut Cochin, Paris, France).

### 4.5. Pre-Processing and Microarray Analyses

The raw microarray data (cel-files) were corrected for background, log_2_-transformed, normalized using the standard method RMA (robust multiarray average) and annotated with the R oligo package 1.42.0 [71]. Expression was summarized at the transcript level. Differential expression testing was conducted by ANOVA after linear model fitting of expression intensities with the limma R package (https://bioconductor.org/packages/release/bioc/html/limma.html, accessed on 27 October 2021). Microarray expression values are represented as log(2) normalized intensities. 

### 4.6. Differentially Expressed Genes and Functional Enrichment Analyses

Differentially expressed genes (DEGs) were filtered with cutoffs of absolute logFC ≥ 0.5 and *p* value ≤ 0.05. Downstream analyses were performed with R version 3.5.0 on a CentOS Linux 7 system (64-bit). Global enrichment analyses were performed using ClusterProfiler, an R package for comparing biological themes among gene clusters [72]. Associations with predicted diseases were performed using EnrichR [39] using a gene set of differentially downregulated common genes in RAD and BPA_RAD conditions. The *XY female disease* and *androgen insensitivity syndrome* gene sets (top-ranked 200 genes linked to these diseases by cooccurrence analyses using the ARCHS4 co-expression matrix and geneRIF datasets) were downloaded from EnrichR’s *Rare Diseases GeneRIF_ARCHS4 Prediction* libraries. Gene expression changes in these two gene sets were evaluated for all treated groups (BPA, RAD and BPA_RAD) and compared to a cohort of unbiased gene sets. For this, we sampled 1000 random sets of 200 genes provided from our unfiltered gene list (DEGs or not) and compared the median distribution of these sets to the median of the *XY female disease* gene set or *androgen insensitivity syndrome* gene set (statistical analyses were performed using a single sample *t*-test). 

### 4.7. Interaction Study on Transcriptome Analyses

An interaction study was performed on log2FC data obtained from fetal and adult transcriptome analyses. These analyses were restricted to annotated transcripts (“Protein coding transcripts”) or performed on all transcripts (“All transcripts”). Types of interaction were interpreted considering that an additive effect occurs when the combined effect of two chemicals is equal to the sum of the effect of each treatment given alone. Potentiation occurs when the combination effect is stronger than the effect of each agent alone. Antagonism occurs when the combined effect is less important than the effect of each agent alone. Only transcripts that were significant differentially expressed in the BPA and/or RAD groups were used for the study. Each interaction type was calculated as follows:$$\mathrm{With}\;r=\left|\frac{logFC\left(BPA\_RADcombination\right)}{logFC\left(BPAalone\right)+LocFC\left(RADalone\right)}\right|$$

The interaction was considered additive when 0.8 ≤ *r* ≤ 1.2, potentiating is considered when *r* ≥ 1.2 and antagonistic when 0.8 ≥ *r*.

### 4.8. Hoechst 33342 and Propidium Iodide Staining and Flow Cytometry

Testis digestion and flow cytometry experiments were performed as previously described [73]. Briefly, seminiferous tubules from 3- and 6- month-old testes of mice, exposed with BPA and/or RAD in utero, were digested with collagenase and trypsin and filtered. A batch of one million cells was diluted in 1 mL of incubation buffer and stained with Hoechst 33342 (5 µg/mL, Sigma-Aldrich, Saint-Quentin-Fallavier, France) for 1 h at 32 °C. Before analysis, propidium iodide was added to exclude dead cells. Analysis was performed using an LSR II cytometer (Becton Dickinson). Spermatogonia and spermatogonial stem cells (side population) were determined according to their capacity of Hoechst exclusion [73]. The ratio of spermatogonia to the number of other germ cell populations (spermatocyte I and II and spermatids) was quantified after analysis.

### 4.9. Sperm Count

Epididymis from 6-month-old mice exposed to BPA and/or RAD mice in utero were mechanically dissociated in 1 mL M2 medium (Sigma–Aldrich) and incubated for 5 min. The sperm released into M2 were diluted at a 1:5 ratio with distilled water and counted manually with a haemocytometer under a microscope.

### 4.10. Testosterone Radioimmunoassay 

Plasma samples from adult mice (3–6 months old) were collected, and their testosterone levels were measured in duplicate by radioimmunoassay, as previously described [74], with [3H] testosterone (NET370001MC, Perkin Elmer). The limit of detection was 80 pg/mL. The intra-assay coefficient of variation (CoV) was 2%, and the inter-assay CoV was 5%.

### 4.11. Partner Preferences and Ultrasonic Vocalization Recordings

Sexual partner preference testing was carried out using the X-maze (Figure 6A). The X-maze apparatus consisted of four arms made of black plastic joined in the middle. The dimensions of each arm were 12 cm (width), 12 cm (high) and 35 cm (length). One of the arms was designated as the starting arm, and the opposite arm was empty. At the ends of the other opposite arms, we placed movable transparent perforated plastics boxes (10 × 10 × 10 cm) containing male mice. These boxes allowed the stimulus animals to move freely but confined them within a small portion of the X-maze. Twenty sexually naive females (2 months old) and 5–7 sexually naive males from each group (CTL, BPA, RAD, BPA_RAD, 6 months old,) were used for the partner preference test. To avoid any supplementary stress during the test, estrus cycle monitoring was performed using vaginal cytology, and female mice were monitored daily for 2 weeks before testing for habituation. Two weeks before the test, female and male mice were housed in the test room. Female mice were habituated to the X-maze for 10 min prior to the beginning of the test. Male mice were placed inside the perforated boxes 10 min prior to the test. All habituation and testing occurred during the dark portion of the reversed day–night cycle and under red light. On the day of the test, estrus cycle was monitored at 9 a.m. and 14 p.m. Estrus females were placed in the starting arm of the maze, and caged males were randomly placed at the end of opposite arms. Exploratory behavior of the female mice was recorded during 5 min, and the time spent by arm was quantified. A preference score was calculated by dividing the time spent investigating one arm with a male (arm A, Figure 6A) minus the time spent investigating the arm containing the other male (arm B, Figure 6A) by the total time spent investigating both compartments. A positive value of the preference score indicates a mate preference directed towards the male in arm A, whereas a negative value indicates a mate preference directed towards the male in arm B. During the test, male ultrasonic vocalizations (USVs) were recorded using a USV-sensitive microphone fixed in front of the box (sampling rate: 250 kHz; FFT-length: 1024 points; 16-bit format; Condenser ultrasound microphone Polaroid/CMPA; Avisoft Bioacoustics, Germany). Recordings were stored on a PC through A/D conversion (Avisoft-UltraSoundGate 416–200, Avisoft Bioacoustics, Germany). Spectrograms were viewed using the R-package, MonitoR [75], and calls were manually detected using the viewSpec function (frequency limits = 20–80 kHz, duration of page length =1 s). Eleven mouse USVs were discriminated (short calls: “short”, “downward”, “upward”, “U-shape”, “chevron” and “flat”; complex and/or harmonic calls: “complex”, “wave”, “harmonic”, “composite”, “one jump” and “multi jump”, Figure 6C) by the shapes of the calls visualized in the spectrogram. As calls from female mice or the other male could be registered, the spectrograms from both tested males recorded during the same test were viewed together (in stereo mode) to exclude calls recorded on both clips. For this reason, the total number of USVs included in this study was lower than the exact USVs emitted by males.

## Figures and Tables

**Figure 1 ijms-22-11808-f001:**
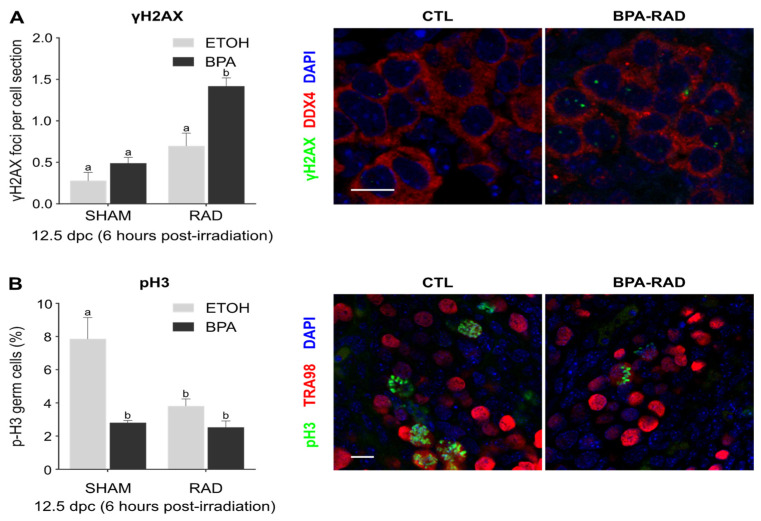
Additive effects of bisphenol A (BPA) and ionizing radiation (RAD) on fetal germ cells. Testes from 12.5 days post coïtum (dpc) embryos exposed to BPA (10 µM) were collected 6 h after irradiation (0.2 Gy), and (**A**) the number of γ-H2AX foci per germ cell (DDX4-positive cells) and (**B**) the percentage of pH3 positive germ cells (TRA98-positive cells) were quantified after immunofluorescence staining. Each bar represents the mean ± sem from 5 independent samples. At least 100 nuclei were counted per sample. Means with different letters are significantly different (*p* < 0.05; two-way ANOVA followed by a Tukey test). Right panels show representative pictures of the staining in control (CTL) and co-exposed (BPA-RAD) gonads. Bars represent 10 µm. (**C**) Gonads from fetal (15.5 and 18.5 dpc) and postnatal (1 day postpartum) mice exposed to BPA and/or RAD were collected, and the number of germ cells per gonad was quantified after immunostaining for TRA98 (left panel). Each point represents the mean ± sem. *n* = 5–7 gonads from independent samples and 3 independent exposures. Means with different letters are significantly different (two-way ANOVA followed by a Tukey test). Right panels show representative pictures of the staining (TRA98 in brown) in 1 dpp control (CTL) and co-exposed (BPA-RAD) testes. Bars represent 100 µm.

**Figure 2 ijms-22-11808-f002:**
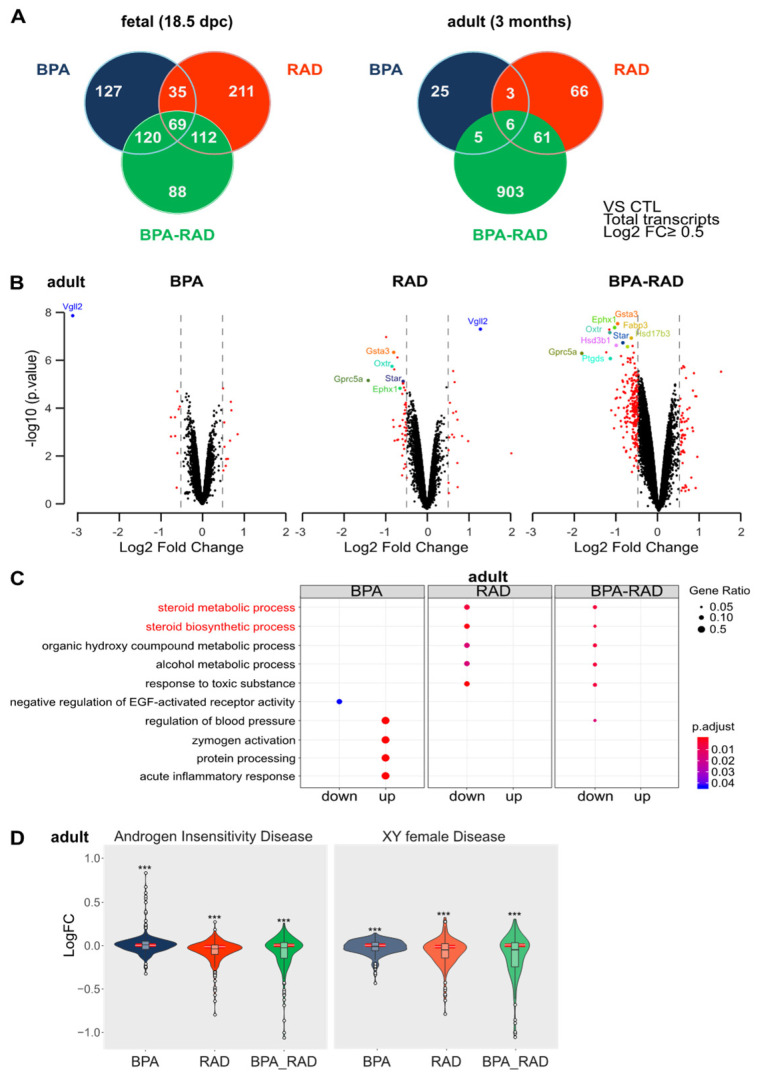
BPA and RAD co-exposure in utero synergistically increase the number of differentially expressed genes (DEGs) in the adult testis. Fetal (18.5 dpc) and adult (3 months) testes from exposed mice (BPA, RAD or combined exposure) or unexposed (CTL as control) mice were collected for transcriptome analyses. (**A**) Venn diagram of DEGs (all transcripts, log2 FC ≥ 0.5) in treated testes vs. control testes. (**B**) Visualization of the DEGs (protein coding mRNA, log2 FC ≥ 0.5) by volcano plots. (**C**) Group comparison of gene ontology biological process categories of significantly downregulated (down) or upregulated (up) genes (protein coding mRNA, Log2 FC ≥ 0.5) between exposed conditions (BPA, RAD or BPA-RAD) and the control (ETOH) group in adult testes. (**D**) Violin plot of LogFC expression of genes included in the gene sets associated with *XY female disease* and *androgen insensitivity syndrome.* Blue, orange or green boxplots represent the interquartile range and median value of the distribution of LogFC in the BPA, RAD and BPA-RAD conditions, respectively. A red rectangle with a white line underneath the box plot represents the interquartile range (red box) and median (white line) of a cohort of unbiased gene sets sampled by simple random sampling of the global gene list. *** *p* value < 0.001 by one-sample *t*-test.

**Figure 3 ijms-22-11808-f003:**
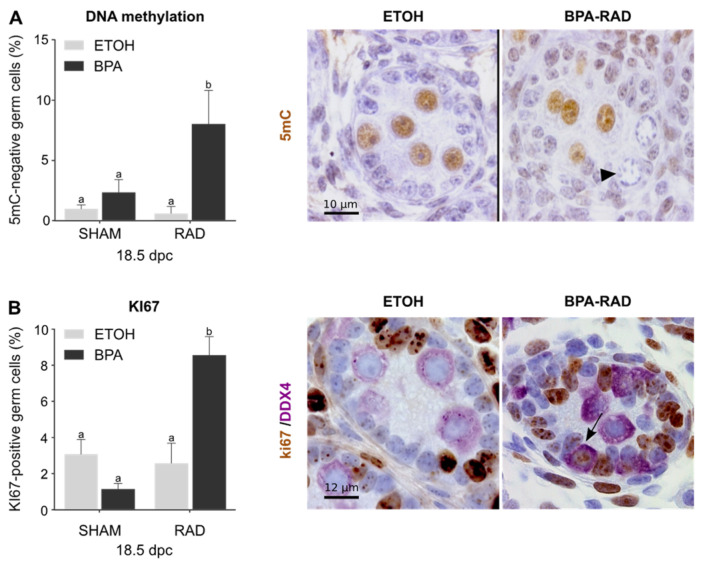
BPA and RAD co-exposure in utero induce gonocyte exit from quiescence. Fetal (18.5 dpc) testes from exposed mice (BPA, RAD or combined exposure) or unexposed (ETOH as control) mice were collected for immunohistological analyses. (**A**) Percentage of 5-methylcytosine (5 mC)-negative germ cells. Each bar represents the mean ± sem from 5–7 independent samples. Means with different letters are significantly different (*p* < 0.05; two-way ANOVA followed by a Tukey test). Right panels show representative pictures of the staining in control (ETOH) and co-exposed (BPA-RAD) gonads. The arrowhead indicates unstained germ cells with typical features of spermatogonia (i.e., proliferative cells and relocation to the basement membrane). Bars represent 10 µm. (**B**) Percentage of KI67 positive germ cells (DDX4 positive cells). Each bar represents the mean ± sem from 5 independent samples. Means with different letters are significantly different (*p* < 0.05; two-way ANOVA followed by a Tukey test). Right panels show representative pictures of the staining of DDX4 (purple) and KI67 (brown) in control (ETOH) and co-exposed (BPA-RAD) gonads. The arrow indicates KI67-positive germ cells.

**Figure 4 ijms-22-11808-f004:**
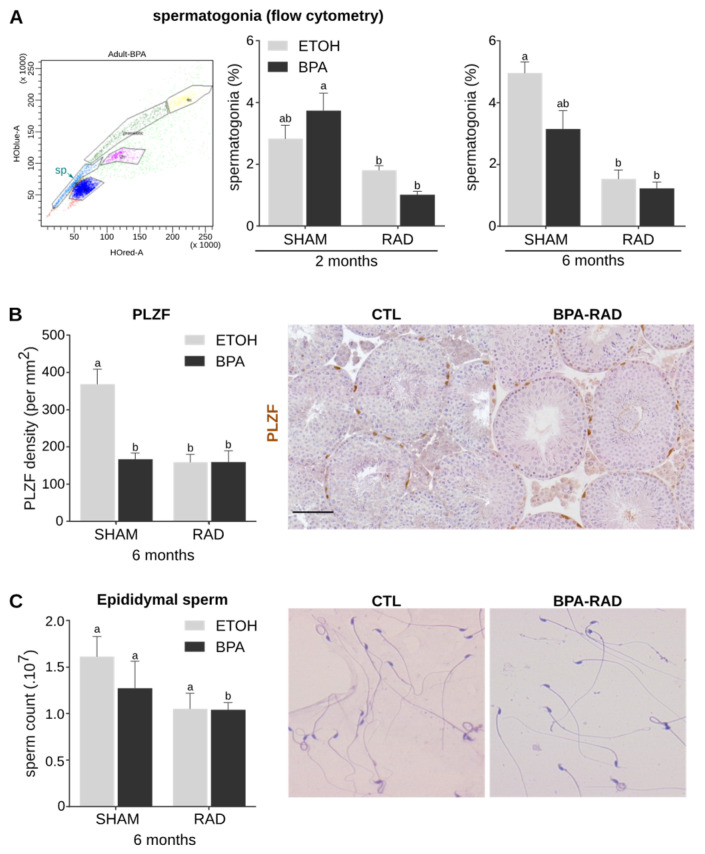
BPA and RAD co-exposure in utero decrease the number of spermatogonia in adult mice. Testes were collected from adult (3-month-old and 6-month-old) in utero exposed mice (BPA, RAD or combined exposure) or unexposed (CTL) mice. (**A**) Percentage of spermatogonia after flow cytometric analyses based on Hoechst and PI fluorescence and light scattering parameters. Left panel: Representative flow cytometry analysis with the side population (sp) corresponding to spermatogonial cells. Middle and right panels: “sp” cell percentage in the total germ cell fraction (sp, premeiotic, 4n, 2n and n germ cells) at 3 months (middle panel) and 6 months (right panel). Each bar represents the mean ± sem from 5–12 independent samples. Means with different letters are significantly different (*p* < 0.05; two-way ANOVA followed by a Tukey test). (**B**) Density of PLZF positive cells per seminiferous tubule area (mm^2^). Each bar represents mean ± sem from 4–5 independent samples. Means with different letters are significantly different (*p* < 0.05; two-way ANOVA followed by a Tukey test). Right panels show representative pictures of the staining in control (CTL) and co-exposure (BPA-RAD) gonads. Bars represent 50 µm. (**C**) Sperm count in the epididymis. Each bar represents the mean ± sem from 4 independent samples. Different letters indicate significant differences between groups (two-way ANOVA, Tukey’s multiple comparisons test, *p* < 0.05). Right panels show representative sperm stained with haematoxylin/eosin (H/E).

**Figure 5 ijms-22-11808-f005:**
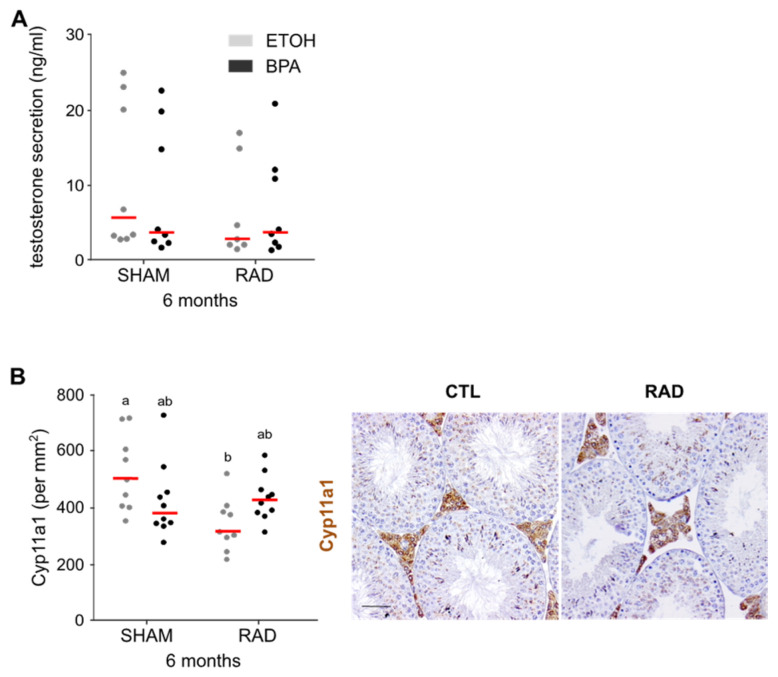
RAD exposure in utero alters steroidogenesis in adult mice. (**A**) Plasma testosterone concentration in control (CTL) and BPA- and/or RAD-exposed adult mice. Each dot represents an individual mouse and medians are represented as horizontal red lines (no statistically significant difference was observed after two-way ANOVA, *n* = 7–8). (**B**) Number of CYP11A1 positive cells per mm^2^ of testis from unexposed and exposed adult mice. Each dot represents an individual mouse and medians are represented as horizontal red lines. Different letters indicate significant differences between groups (two-way ANOVA, Tukey’s multiple comparisons test, *p* < 0.05, *n* = 9–10). Right panels show representative pictures of the staining in control (CTL) and RAD-exposed (RAD) gonads. Bars represent: 50 μm.

**Figure 6 ijms-22-11808-f006:**
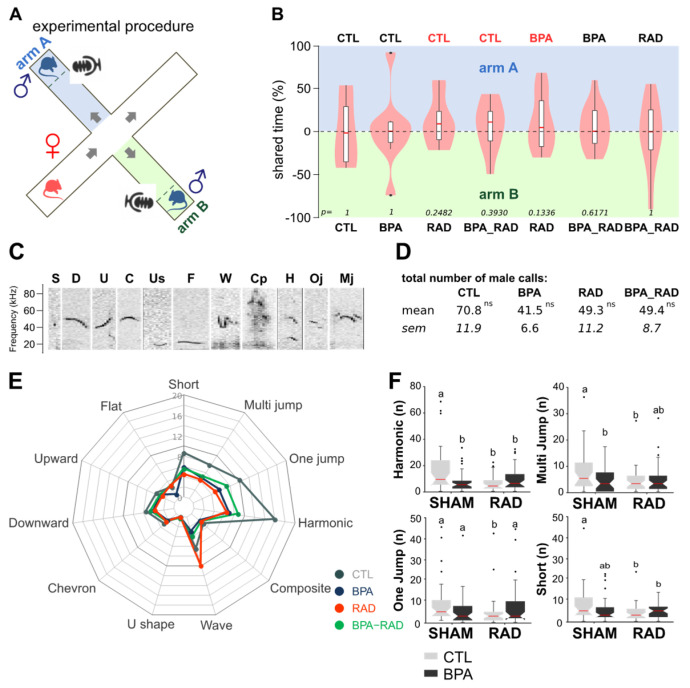
RAD exposure in utero alters the emission of male mouse ultrasonic vocalizations during courtship. (**A**) Schematic diagram of the partner preference test design. A sexually receptive free-moving female was placed in an arm of the X-maze. Two adult males from each experimental group (CTL, BPA, RAD, BPA_RAD) were caged in one arm, and the time spent in each arm was measured. At the same time, male ultrasonic vocalizations (USVs) were recorded by microphones placed close to the males. (**B**) Violin plot of the female time spent in the two arms (% of time spent in arm A minus % of time spent in arm B, chi-square test, *n* = 5–7). (**C**) Typical spectrograms illustrating eleven distinct call categories. (**D**) Total number of USVs emitted by control (CTL), BPA and/or RAD exposed adult mice (data are expressed as the means ± sem, no statistically significant difference was observed after two-way ANOVA). (**E**) Radar chart of the total number of syllables emitted by control (CTL)–, BPA– and/or RAD-exposed adult mice (data are expressed as the means). (**F**) Notched box plot of the total number of harmonic one-jump, multiple jump and short calls. Different letters indicate significant differences between groups (two-Way ANOVA, Tukey’s multiple comparisons test, *p* < 0.05, *n* = 30–40).

**Table 1 ijms-22-11808-t001:** Type of interaction between BPA and RAD on gene expression. Analyses were performed on a set of differentially expressed gene transcripts (all transcripts) and on a set of protein coding transcripts.

	Number of Transcript *	Type of Interaction (% of Misregulated Transcripts)			
		Potentiation	Additivity	Antagonism	Unexpected
Fetal					
All transcripts	762	9.8	26.6	57.9	5.7
Protein coding mRNA	238	18.1	7.1	64.3	10.5
Adult					
All transcripts	1069	71.2	16.3	11.2	1.2
Protein coding mRNA	308	59.7	26.9	12.3	1.0

* Transcripts were obtained from DEGs (LogFC ≥ 0.5).

## Data Availability

Datasets are available at the European Bioinformatics Institute under accession n° (E-MTAB-10982). The authors declare that the data supporting the findings of this study and custom code generated for the manuscript are available from the Lead Contact upon request.

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
