# Peer review of "Unexpected Interacting Effects of Physical (Radiation) and Chemical (Bisphenol A) Treatments on Male Reproductive Functions in Mice"

_ijms, 2021, doi:10.3390/ijms222111808_

Round 1

Reviewer 1 Report

Authors report the results of in vitro (HeLa cells) and in vivo (mice) to determine the effect of foetal exposure to two environmental toxicants (bisphenol and radiation) and the repercussions on male reproduction, by studying various parameters along of the developmental stages of the mice. This study focuses a current problem and the results deserve to be published. This work is a very complete study, with clear results that are well documented with appropriate figures and tables. The methods have the needed detail and are very well written.

Minor correction:

Table 1 (last column) replace 1,2 by 1.2

Figure 1C, in the xx axis legend, replace jpc by dpc and jpp by dpp.

Supplementary material is fine

Author Response

Response to Reviewer 1 Comments_Minor correction:

"Table 1 (last column) replace 1,2 by 1.2 : 

Figure 1C, in the xx axis legend, replace jpc by dpc and jpp by dpp."

Thank you for the comments.  All corrections are done directly in the text and the figure and are highlighted in yellow.

Reviewer 2 Report

In this manuscript, authors investigate the effect of combined foetal exposure to BPA and g-ray on male reproduction. The proliferation and transcription of germ cell in testis are affected after BPA and g-ray exposure. The work was presented well except a few minor exceptions.

  1. Besides description in the materials and methods section, it is also recommended to mention in the results section briefly that how those mouse models were established. For example, how long does the mice were exposed to g-ray? The dose for g-ray and BPA?
  2. Please use labels for statistical difference consistently in figures and texts. a, b or *p, **p
  3. Before fertilization, sperm have to undergo several physiological processes including spermatogenesis, maturation in epididymis, functional regulation such as capacitation in female reproduction tract. Besides the effect on testicular sperm, have authors explored any potential effects on sperm maturation or later functional regulation, especially considering it is already known that CatSper, sperm-specific cation channel, could be affected by BPA to regulate sperm function?
  4. In vivo or in vitro fertilization assays are a comprehensive technique to evaluate sperm function or male fertility. Have authors done fertility test, such as IVF, litter size counting?

Author Response

Response to Reviewer 2 Comments_Minor correction:

- Point 1 : " Besides description in the materials and methods section, it is also recommended to mention in the results section briefly that how those mouse models were established. For example, how long does the mice were exposed to g-ray? The dose for g-ray and BPA? "

Response 1 : We are agree with this comment, we have added in the introduction the following text (line 91-96) : Four experimental groups were studied: nonirradiated and non BPA-exposed control mice (CTL); nonirradiated and BPA-exposed mice (BPA); irradiated and non-BPA-exposed mice (RAD) and irradiated and BPA-exposed mice (BPA_RAD). For BPA exposure, pregnant mice were exposed to 10 µM BPA (corresponding to 0.5 mg/kg/day) in drinking water from 10.5 dpc to 18.5 dpc. For RAD exposure, 12.5 dpc pregnant mice were irradiated with a single whole-body dose (0.2 Gy). »

- Point 2 : "Please use labels for statistical difference consistently in figures and texts. a, b or *p, **p".

Response 2 : Consequently to this comment, we have modified the Figure1.

-Points 3 and 4 : "Before fertilization, sperm have to undergo several physiological processes including spermatogenesis, maturation in epididymis, functional regulation such as capacitation in female reproduction tract. Besides the effect on testicular sperm, have authors explored any potential effects on sperm maturation or later functional regulation, especially considering it is already known that CatSper, sperm-specific cation channel, could be affected by BPA to regulate sperm function? "

"In vivo or in vitro fertilization assays are a comprehensive technique to evaluate sperm function or male fertility. Have authors done fertility test, such as IVF, litter size counting? "

Response 3 and 4 : After in vitro capacitation by albumin treatment, we have monitored the sperm motility in all conditions. We did not observed any difference between conditions. In addition, we have counted the cumulative litter size between condition (n=5 couples by groups, one CTL/BPA/RAD or BPA-RAD male mate with CTL female during 350 days).

As you can see, we did not observed any difference in terms of cumulative litter size and latency between litters. The fact that we did not observe any difference between conditions despite the decrease of sperm product observed in BPA_RAD condition was not a surprise. In rodents, it is well known that a decrease in spermatozoa quality and/or quantity may have no consequence on male fertility. As an example, it was observed that a reduction by 55% in spermatozoa production does not impact male fertility in mice (Forand et al, 2009). These data only suggest that in utero treatment with BPA and/or RAD did not modify the epididymal sperm maturation and have only consequences on the sperm product and SSC pool. As suggested by the reviewer 2, we have added this supplementary Figure (Supplementary Figure 5) linked with this comment (line 265-267): “No modification of the litter size was observed between conditions suggesting that BPA and/or RAD have no impact on sperm maturation (Supplementary Figure 5).”
